# Evaluation of the Cell Concentration in Suspensions of Human Leukocytes by Ultrasound Imaging: The Influence of Size Dispersion and Cell Type

**DOI:** 10.3390/s23020977

**Published:** 2023-01-14

**Authors:** Luis Elvira, Alba Fernández, Lucía León, Alberto Ibáñez, Montserrat Parrilla, Óscar Martínez, Javier Jiménez

**Affiliations:** 1Instituto de Tecnologías Físicas y de la Información (CSIC), Serrano 144, 28006 Madrid, Spain; 2Newborn Solutions, Baldiri Reixac, 4-12 i 15, 08028 Barcelona, Spain

**Keywords:** concentration assessment, cell suspension, ultrasound imaging

## Abstract

This work focuses on the use of ultrasound imaging to evaluate the cell concentration of dilute leukocyte suspensions in the range of 10–3000 cells/µL. First, numerical simulations were used to study the influence of the size dispersion and the leukocyte type on the performance of the concentration estimation algorithms, which were developed in previous works assuming single-sized scatterers. From this analysis, corrections to the mentioned algorithms were proposed and then the performance of these corrections was evaluated from experiments. For this, ultrasound images were captured from suspensions of lymphocytes, granulocytes, and their mixtures. These images were obtained using a 20 MHz single-channel scanning system. Results confirmed that concentration estimates provided by conventional algorithms were affected by the size dispersion of cells, leading to a remarkable underestimation of results. The proposed correction to compensate for cell size dispersion obtained from simulations improved the concentration estimation of these algorithms, for the cell suspensions tested, approaching the results to the reference optical characterization. Moreover, it was shown that these models provided a total leukocyte concentration from the ultrasound images which was independent of the relative populations of different white blood cell types.

## 1. Introduction

Cell concentration in tissues and fluids constitutes a metric of primary importance to assess the health state of living organisms. While the measurement of the cell population in blood is the most common concentration assessment performed in the clinical practice, cell concentration evaluation in other fluids such as urine and serous fluids also provide information about relevant pathologies, as it is the case for infections (for example, meningitis [1,2]), cancer [2,3], and internal hemorrhages [2]. Usually, the estimation of cell concentration is made ex vivo; however, recently, non-invasive methodologies such as confocal microscopy and optical coherence are attracting interest for their advantages as rapid and innocuous tools able to perform this task in vivo [4,5]. Ultrasound, which is able to penetrate deeper in the organism than optic technologies, also arises as a potential method to assess cell concentration non-invasively. This work was motivated by such purpose and aims to develop algorithms to evaluate cell concentration suitable for in vivo applications

Ultrasound backscattering data were successfully used to estimate the concentration of different cell suspensions: Chen et al. [6] evaluated erythrocyte concentrations (104–106 cells/μL) and Jimenez et al. [7] analysed leukocyte concentrations (5–60 cells/μL) using the power spectrum of backscatter signals in the samples. This strategy, which was also used to analyse other properties of cell suspensions, such as cell aggregation [8], requires the knowledge of the relation between the intensity reaching the targets and a reference intensity, which is often achieved through a previous calibration of the system. The methodology was simple and well fitted for ex vivo samples where this intensity reference can be characterized. Dynamic light scattering emerged as a powerful technique to analyse concentration and size of particles in suspension, being applied successfully to analyse the concentration and size of small cells such as some bacteria (0.5–2 μm size) [9] and platelets (2–3 μm size) [10]. The technique makes use of Brownian motion to relate the change of successive acquired signals to the particle size. Thus, the ultrasound-based version of this technique, dynamic ultrasound scattering, arose as a technique able to analyze both the scatterers’ concentrations and sizes without alternative measurements [11,12]. This could be of great interest when the sizes of cells are unknown and/or present a wide variability. For the case of larger scatterers, where Brownian motion could not be used because of its small influence, particle sedimentation under stable and controlled conditions was proposed as a method to relate size and particle movement [12].

When using ultrasonography for in vivo imaging in the clinical setting, different tissues are crossed by ultrasound to reach the region of interest. It is known that these tissues degrade the quality of the obtained images [13]. Moreover, these tissues make the ultrasound energy which reaches the region of interest variable among different patients. As a result, a suitable in vivo methodology to extract the cell concentration information from ultrasound images should be independent of the amplitude of the waves reaching the cells. In addition, such methodology should also be efficient for those concentration ranges relevant to diagnostic thresholds (for diagnostic and screening purposes) and/or above these levels (to monitor the effects of the medical treatment on the cell concentration). Finally, imaging-based technologies provide a well-fitted strategy to localize adequate regions of interest to perform the required concentration measurement, despite the varying anatomical characteristics of different patients.

The present work is focused on the ultrasound-based concentration assessment of leukocytes in serous body fluids (cerebrospinal fluid (CSF), pleural fluid, and lacrimal fluid, etc.) where very low concentrations of cells are expected in healthy conditions. The appearance of an abnormal leukocyte concentration in such fluids is a symptom of an infection or an inflammatory process. The concentration range of interest depends on the particular fluid analysed. For example, leukocyte concentrations above 20 cells/µL in CSF can be correlated with meningitis in neonates [14], and severe cases of bacterial meningitis give rise to concentrations above 1000 leukocytes/µL. This means that concentration measurements within these two orders of magnitude would be relevant for monitoring this pathology.

In the last years, several algorithms were developed to assess the concentration of cells independently of the ultrasound amplitude reaching them. Tunis et al. [15] evaluated concentrations of acute myeloid leukemia cells (12.5 μm diameter) in the range of 5 × 102–104 cells/μL using envelope statistics. Lee et al. [16] proposed a technique which was successfully applied to T-cells suspensions in the range of 1–20 cells/μL, and to silica and polystyrene (PS) microparticles (10 and 15 μm diameter) in the range of 1–200 part/μL. Elvira et al. [17] proposed a method based on the detection and quantification of ultrasound echo patterns from scatterers (EPQ) which was applied under poor signal to noise ratio (SNR) conditions to PS microparticles (7 μm and 12 μm diameter) in the 5–100 part/μL range. Fernández et al. [18] used another algorithm based on signal envelope statistics analysis (ES) to assess the concentration of PS microparticles (7 μm and 12 μm diameter) in a wider range of 5–3000 part/μL. In these last two works it was shown that concentration estimations provided by these models were affected by the size dispersion of the scatterers. The EPQ model proved to have a higher robustness against noise (acoustical or electrical) than the ES model. However, the applicability concentration range of the EPQ method was limited (<100 cells/µL) and lost efficiency when echo overlapping increased even before this limit. On the contrary, the ES algorithm was advantageous when dealing with echo overlapping and improved its robustness against noise for high concentrations [19]. As a result, it was proposed that when the EPQ model provided concentrations below 50 cells/µL (0.03 cells/resolution cell), this value could be assumed as the actual result. For those concentrations higher than this threshold, the ES algorithm was applied to assess the final value.

The present work further develops and analyses the performance of both the EPQ and ES ultrasound-based methodologies in human cells suspensions, in which size variations are expected. In particular, human leukocyte suspensions, with cell concentrations in the range of 10–3000 cells/μL, which would be relevant for meningitis screening, were characterized. To this end, first of all, the influence of the cell size dispersion on the concentration estimations is analysed by means of numerical simulations. From them an algorithm to compensate for the effect of size dispersion of the cells is proposed. Finally, experimental measurements of cell concentration in suspensions constituted by different leukocyte types (granulocytes and lymphocytes) are presented and discussed. Granulocytes and lymphocytes were chosen for this study as their sum constitutes for more than 90% of the total leukocytes. There are different types of granulocytes: neutrophils, eosinophils, and basophils, and, among them, neutrophils are predominant.

## 2. Materials and Methods

### 2.1. Methodology to Compensate the Size Dispersion of Scatterers in the Concentration Estimation Algorithms

The methodology to assess cell concentration from ultrasound images followed in the present paper is based on the models proposed by Elvira et al. [17], Fernández et al. [18], and Fernández [19]. The first model detects and quantifies patterns of ultrasound echoes coming from individual scatterers [17,19]. The algorithm needs to be tuned to the ultrasound device and configuration used, which is achieved by means of empirical calibration or numerical simulation. Then, the algorithm provides an absolute cell concentration assessment and, when applied to cell concentrations in the range of 5–100 cells/µL, proved to be robust against amplitude variability in part arising from the tissue attenuation that is different for each patient. For the ultrasound system used (20 MHz focalized single channel scanning transducer), this corresponds to 0.003–0.06 cells/resolution cell; the resolution cell is defined as the volume where the ultrasound echo from a given cell decreases down to 3 dB below the maximum amplitude (cell at the focal point).

The second model uses envelope statistics to estimate scatterer concentration [18,19]. This model showed good performance over a wider concentration range (0.003–1.8 cells/resolution cell) which corresponds to 5–3000 cells/µL. In particular, the work proposed the homodyned K distribution to model cell backscattering. The homodyned K distribution is a multi-parametric function able to describe the distribution of amplitudes of the ultrasound signal envelope backscattered by particles or cells dispersed in a fluid. Other commonly used functions such as the Rayleigh distribution, the Rice distribution, the K distribution, and the Nakagami distribution, can be obtained as particular cases of the homodyned K distribution, which can be defined with the following integral expression:(1)$$p_{A}\left(A\right)=A\int_{u=0}^{\infty }u\frac{J_{0}\left(us\right)J_{0}\left(uA\right)}{{\left(1+\frac{u^{2}\sigma^{2}}{2\mu }\right)}^{\mu }}\;du$$
where A is the amplitude of the signal envelope, *s*, *σ*, and *μ* are the three parameters characterizing the coherence component, the variance, and the number of scatterers, respectively. These parameters define the shape of the probability distribution, pA(A), of the amplitudes. In particular, the *μ* parameter showed a good correlation with the scatterer concentration. Field simulations are used to obtain the relation between the *μ* parameter and the absolute concentration, since such relation depends on the ultrasound device and the pulse configuration used. These simulations were performed in MATLAB using the frequency domain model proposed in [17,19]. Once the relation between the *μ* parameter and the absolute concentration is obtained, the method is independent of the signal amplitude reaching the scatterers, which makes it suitable for analysing in vivo images.

Assuming Rayleigh scattering conditions (weakly scattering particles and particle sizes smaller than the wavelength [8]), both the quantification of cell echo patterns and the envelope statistics models do not depend on the particular size of the cells in monodisperse suspensions. However, it was shown [17,18,19] that the size dispersion may have an impact on the estimates provided by the algorithms and, as a consequence, corrections from these effects should be included in the models. To this end, the following methodology was followed in the present work:The size distribution of different types of leukocytes (lymphocytes and granulocytes) was characterized by optical microscopy.Backscattering field simulations were performed for cell suspensions showing a Gaussian size distribution with the central diameters and standard deviations measured for both lymphocytes and granulocytes. Simulations were made using the spectral acoustic field model described in [17,19] and concentrations in the range of 10–1000 cells/µL.Echo pattern quantification (EPQ) and envelope statistics (ES) algorithms developed for single-size scatterers were applied to these simulations for concentrations below 100 cells/µL and for the whole simulated range, respectively.Size dispersion correction factors as a function of the cell concentration were obtained for both models.A single algorithm to estimate the concentration of leukocytes from ultrasound images was implemented. This algorithm combined both EPQ and ES models, incorporating the aforementioned factors to compensate for the size dispersion.The performance of this algorithm was evaluated using experimental ultrasound images obtained from leukocyte suspensions of different types and concentrations.

### 2.2. Experimental Methodology

#### 2.2.1. Preparation of Cell Suspensions

The cell suspensions analysed in the present study were made using lymphocytes and granulocytes purified from human blood samples and suspended in PBS (Phosphate Buffer Saline) medium. Lymphocytes are mononuclear leukocytes constituting about 20–45% of total leukocytes in human peripheral blood [20]. According to the same reference, granulocytes or polymorphonuclear (PMN) leukocytes represent about 40–75% of total leukocytes, and both granulocytes and lymphocytes constitute more than 90% of total white blood cells.

Leukocyte samples were obtained by the Myeloid Cell Biology Group of the CIB (Centro de Investigaciones Biológicas), CSIC (Consejo Superior de Investigaciones Científicas), from healthy blood donors at the Centro de Transfusión (Madrid, Spain), which were previously anonymized by the Comunidad de Madrid Blood Bank. Ethical approvals for all blood sources and processes used in this study were approved by the CIB Ethics Committee. All experiments were carried out in accordance with the approved guidelines and regulations.

After applying a density gradient medium [21], mononuclear cells were separated from granulocytes and most of the erythrocytes. To obtain a purified lymphocyte suspension from the vial containing mononuclear cells, monocytes were extracted by magnetic cell sorting using human CD14 microbeads (#130-050-201, Miltenyi Biotech, Bergisch Gladbach, Germany). After this, the lymphocyte sample was treated with an ACK buffer (made of ammonium chloride and potassium bicarbonate) to lyse the remaining erythrocytes. For that, 1 mL of the cell suspension was added to 10 mL of ACK buffer in a Falcon tube. The tube was inverted three times and left to rest for 3–5 min. Then, the sample was centrifuged (300× *g*) for 5 min at ambient temperature. Supernatant was extracted and discarded, and the cells were resuspended in 5 mL of PBS. This suspension was centrifuged again (300× *g*, at 2–8 °C), and the supernatant was extracted and discarded. A 1 mL sample of PBS was then added to the suspension and, finally, the lymphocyte concentration was evaluated by the microscope.

To obtain the granulocyte purified sample, a separation process to decant the erythrocytes was conducted using a Dextran solution, based on the protocol described by García et al. [22]. For this, two Falcon tubes, each with 2 mL of Dextran 6% in PBS and 10 mL of the sample containing RBC and granulocytes, were prepared. Both tubes were inverted three times and left to rest for 45 min. The clearer upper layers of both tubes containing granulocytes were extracted and mixed into another Falcon tube, which was centrifuged (300× *g*) for 10 min at ambient temperature. The supernatant layer was extracted, and the cells were softly mixed with 1 mL PBS. Then, lysis of erythrocytes still remaining in the suspension of granulocytes was achieved following the same procedure followed to purify the suspension of lymphocytes. At the end of the process, the concentration of granulocytes was evaluated by the microscope. Following Carr and Rodak [23], this sample was constituted mainly of neutrophils as they represent about 90% of the whole granulocyte content in human blood.

From the lymphocyte and granulocyte purified suspensions, further dilutions in PBS were made to obtain a 4 mL minimum volume of 30, 100, 300, 1000, and 3000 cells/µL. The cells of these suspension series were either granulocytes, lymphocytes, or a mixture of them. Regarding these mixtures, relative concentrations of 20% lymphocytes/80% granulocytes, 50% lymphocytes/50% granulocytes, and 80% lymphocytes/20% granulocytes were analysed in this work.

#### 2.2.2. Suspension Characterization by Microscopy

Cell suspensions were characterized by optical microscopy (Eclipse 50i, Olympus, Tokyo, Japan) to obtain their concentration and size distribution data. The cell concentration of each sample was analysed immediately after the acquisition of a set of ultrasound images to diminish differences in the sample concentration caused by deposition, adhesion to the tube walls, or cell lysis between the ultrasound and the optical images. Higher concentrations (>100 cells/µL) were assessed using a Neubauer chamber (0.1 µL/sample) and the rest were assessed using a Fuchs-Rosenthal chamber (3.2 µL/sample). Two samples were evaluated from each concentration analysed.

To evaluate the cell size distribution, photographs of most concentrated solutions (around 3000 cells/µL) of lymphocytes and granulocytes were taken before further mixtures and dilutions were made. These images were analysed using the software ImageJ. For these analyses, first, the length scale was defined, and the contrast of the images was enhanced, and thresholds were created to improve particle recognition. Then, the “Analyze particles” function was used to generate a vector with the areas of all the cells detected in the image. Finally, using the software Origin, these areas were converted to diameters assuming circular cells and cell histograms were obtained from the corresponding diameter distributions.

#### 2.2.3. Suspension Characterization by Ultrasound Imaging

Ultrasound acquisitions were obtained using a focalized transducer from Imasonic (Voray sur l’Ognon, France), working at a central frequency of 20 MHz, and showing a 13 MHz bandwidth (−6 dB). The transducer has a 7 mm diameter and 14 mm focal distance. It was put in contact with the sample through water, and a polyethylene film closed the tube that contained the sample to be measured. Regarding the electronics used for the emission and reception of pulses, the Difrascope system (Dasel Sistemas SL, Arganda del Rey, Spain) was used. Square single pulses of −30 V were emitted and 8 acquisitions of 400 samples, using a 100 MHz sampling frequency, were obtained for each measuring point. This group of 8 acquisitions, captured with a frequency rate of 15 kHz, were averaged to reduce noise, providing the resulting A-scan for this point.

The transducer was attached to a 3D positioning system consisting of three moving stages (model Q-521.330, PI, Karlsruhe, Germany) which allowed the correct alignment and coupling with the samples. One of the axes was used for driving linear back and forth runs 12 mm long. The step between the acquisition of two successive A-scans throughout the run was set to 16 µm. A B-scan scan was acquired and stored every 3 s, and a total set of 10 images were recorded for each sample. Following this protocol, the leukocyte concentration estimation of one sample can be performed in less than a minute as the time needed for the assessment of cell concentration in a conventional computer took about 10 s.

## 3. Results

### 3.1. Influence of Size Dispersion on Model Predictions

Following the methodology described in Section 2.1, the first step was to obtain the size distribution of the two types of leukocytes (lymphocytes and granulocytes) under study, which were experimentally characterized by optical microscopy. Morphologically, lymphocytes are round in shape with mononuclear leukocytes being, in general, smaller than the rest of the leukocytes. On the other hand, granulocytes are commonly larger and more dispersed in size than lymphocytes; they contain several nuclei and present irregular borders. Figure 1 shows lymphocytes (1.a) and granulocytes (1.b) in the Neubauer plate at cell concentrations of around 3000 cells/µL. From the images of cells acquired using the microscope, size histograms were obtained for each kind of cell. An example of two of these histograms with a Gaussian fitting superimposed is represented in Figure 1c,d for lymphocytes and granulocytes, respectively. Averaged mean diameters and standard deviations of the lymphocytes and granulocytes analysed in this study are shown in Table 1.

Then, 10 simulated ultrasound images for five different concentrations (10, 30, 100, 300, and 1000 cells/µL) and for each cell type were computed using the spectral acoustic field model described in [17,19]. This gave a total of 100 simulations. Next, echo pattern quantification (EPQ) and envelope statistics (ES) models developed for single-sized scatterers were applied to these images for estimating cell concentration. Results are shown in Figure 2 for both models and for both cell types. Symbols correspond to the average of the 10 estimations for each concentration and cell type, and the error bars were obtained from the standard deviations of the 10 simulations. Given the applicability ranges suited for the models, the EPQ model was not applied for concentrations above 100 cells/µL.

These plots show that similar results were provided for both types of cells. It was also observed that both models underestimate the cell concentration of simulated images as a result of the scatterer size dispersion, with the ES model giving a more pronounced underestimation than the EPS model. Finally, both maintained a growing dependence from the cell concentration along the applicability ranges with no local maximum. This is an important feature because the presence of a local maximum would have implied an uncertainty of the actual cell concentration calculated from the ultrasound estimation because, in such a case, for a given ultrasound estimation, at least two possible cell concentrations could be obtained.

The relationship between the concentration simulated and the concentration calculated by the models may be used to obtain correction functions to compensate for the observed deviation. Then, the compensated concentration *C_ci_* (*i*∈{EPQ, ES}) using the model for a given suspension may be obtained by multiplying the estimation *C_i_*, provided by that model for single-sized scatterers, by a correction factor
(2)$$\{\begin{array}{c} C_{cEPQ}=C_{EPQ}\;\times \;F_{EPQ}\left(C_{EPQ}\right)\\ C_{cES}=C_{ES}\times F_{ES}\left(C_{ES}\right) \end{array}$$

From the values plotted in Figure 2, the compensation factors vs. first concentration estimation can be calculated to obtain the needed correction for any concentration in the range of 0–100 cells/µL for the EPQ model and in the range of 0–1000 cells/µL for the ES model. Considering the small differences shown in Figure 2 for both leukocyte types simulated, only one compensation function was proposed for each algorithm, fitting this function to the combined values given by granulocytes and lymphocytes. These compensation functions were fitted by:(3)$$F_{EPQ}=0.049\;C_{EPQ}+0.43\;\mathrm{EPQ}\;\mathrm{model}$$
(4)$$F_{ES}=\frac{8.6C_{ES}}{19.8+C_{ES}\left(1+\frac{C_{ES}}{215}\right)}\;\;\mathrm{ES}\;\mathrm{model}$$
and are plotted in Figure 3.

### 3.2. Proposed Algorithm for the Evaluation of Cell Concentration in Suspensions with Size Dispersion

Following Fernández [19], a general mixed model is also proposed in the present work that combines the quantification of the echo pattern and the statistics of the envelope model. As mentioned in Section 1, Fernández [19] proposed that when the EPQ model provided concentrations below 50 cells/µL (0.03 cells/resolution cell), the concentration given by the EPQ model could be assumed as the actual result. For those concentrations higher than the 50 cells/µL threshold, the ES algorithm was applied to assess the final value. To make this threshold work for leukocyte suspensions, and according to Equations (2) and (3), the actual concentration $\;C_{cEPQ}=$ 50 cells/µL was obtained when $C_{EPQ}=$ 28 cells/µL. As a result, when the concentration estimated by the EPQ model was higher than 28 cells/µL, the ES model was used instead,
(5)$$\begin{array}{c} \begin{array}{cc} C=\bar{C_{j}} & j\left[1,\;10\right] \end{array}\\ C_{j}=\{\begin{array}{c} \begin{array}{cc} C_{cEPQ} & C_{EPQ}\leq 28\frac{\mathrm{cel}}{\mu L} \end{array}\\ \begin{array}{cc} C_{cES} & C_{EPQ}>28\frac{\mathrm{cel}}{\mu L} \end{array}\; \end{array} \end{array}$$
where $\bar{C_{j}}\;$corresponds to the average of the 10 concentrations.

### 3.3. Cell Concentration Assessment in Suspensions of Lymphocytes and Granulocytes

The leukocyte population in serous body fluids is a mixture of different types of cells. However, the relative concentrations of different white blood cells are unknown a priori and depend on each individual and the state of his/her health. These relative concentrations may affect the global standard deviation of leukocyte sizes as a result of the mixture and, as a consequence, it may affect the concentration estimate. However, assuming that the change of the standard deviation of cell sizes was moderate enough, the same correction factors obtained previously for single-cell suspensions (lymphocytes or granulocytes) may be used for their mixtures. In the following section, this hypothesis is evaluated.

As an example, some of the images captured with different cell concentrations are shown in Figure 4. The transducer focalization gives rise to the concentration of echoes at the central part of the images, where the focus was placed.

Considering that diameters of both lymphocytes and granulocytes are significantly lower than the wavelength used in this work (75 µm), it was expected that no remarkable difference could be seen in the ultrasound images obtained from them, as is shown in Figure 4. The size and intensity of the echoes depend on the pulse length, the position of scatterers relative to the focus, and the overlapping of echoes coming from different nearby cells. Sometimes strong echoes appear in the images (see arrows at Figure 4a), which may correspond to impurities in the sample. However, the amount of these echoes is much lower than the amount of cell echoes and quantification algorithms may correct their influence. Other typical image disturbances such as “clouds of noise” occasionally appear (yellow circle of Figure 4c), which may be related to mechanical decoupling problems between the emitter (transducer with water column) and the polyethylene film covering the tubes with the samples.

Estimates of cell concentrations obtained from the ultrasound imaging of suspensions formed by three different mixtures of lymphocytes and granulocytes are shown in Figure 5, merged with results of pure granulocytes and lymphocytes suspensions. Considering that 5 different dilutions were obtained from each type of suspension, a total of 25 different samples were evaluated. Figure 5a shows the estimates obtained using the proposed algorithm without any compensation of size dispersion. According to our expectations, there were no relevant differences between the samples with a predominance of lymphocytes from those with a predominance of granulocytes. It can be seen that most of the concentrations estimated remained below the microscope counts marked by the diagonal black line. Linear fitting in the log scale considering all samples was obtained and plotted in grey, showing a 0.75 slope, confirmed a bias in relation to the reference optical count marked by the black line.

Figure 5b shows the same set of estimates after applying the compensation factors obtained from the simulations. As before, there are no significant differences in the behavior of the estimates for different cell types. However, the compensation made the concentration estimates better fit the values obtained by microscope count, and the linear fit (grey) became closer to the black line, corresponding to the identity between ultrasound and optic assessment, with a slope of 0.91, much closer to its ideal value of one.

Figure 5 shows that dispersion size compensation was needed to assess the concentration of leukocytes in suspension, and that the methodology proposed based on simulations can be applied to cell suspensions of actual human cells. Moreover, the analysis of ultrasound images obtained from mixtures of lymphocytes and granulocytes gave an estimation of the total cell concentration and was not affected by their relative concentrations. These results are aligned with previous findings of Jimenez et al. [7] obtained for more restricted conditions (concentration evaluation method based on backscattering power which could only be used when calibration was possible, and in the 0–60 cells/µL concentration range) where it was also found that cell concentration estimations were not affected by differences of the relative populations of lymphocytes and polymorphonuclear (PMN) cells.

## 4. Conclusions

It was shown that the size dispersion of cell populations has a noticeable impact on the concentrations estimated from ultrasound images. This is due to the fact that theoretical models to perform this task are commonly developed assuming single-sized acoustic scatterers. In this work, a methodology to correct these algorithms for being applied to the evaluation of cell suspensions was proposed and tested. Ultrasound images, using a scanning single-channel focalized transducer working at 20 MHz, were obtained from different white blood cell suspensions. These suspensions included lymphocyte suspensions, granulocyte suspensions, and mixtures of both kind of cells in the range of 10–3000 cells/µL.

The proposed compensation for cell size dispersion was applied to the models, which was able to improve the concentration assessment of white blood cells in suspension in the mentioned concentration range. The results obtained in mixtures of lymphocytes and granulocytes showed that no bias related to a hypothetical predominant detection of granulocytes nor lymphocytes using ultrasound was found at the frequency used in this work. Numerical methods used to estimate the cell concentration were robust against the relative concentration of both kind of cells and, as a consequence, these models provided a total leukocyte concentration from the ultrasound images which was independent of these relative populations. It is expected than the presence of other leukocytes not included in this study as monocytes would not be relevant because of their low concentration relative to the other white blood cells. The size distribution of leukocytes deserves a deeper analysis to assess, on one hand, the potential influence of the variability in the size and dispersion of leukocytes belonging to different patients. On the other hand, other possible alterations caused by the experimental technique used in this work to obtain cell suspensions might be evaluated. This way, a more extensive in vivo and ex vivo analysis of recently extracted leukocyte samples would be desirable to allow a fine tuning of leukocyte concentration estimators developed in this study. In real in vivo settings, the influence of tissues on the concentration estimation, due to losses in signal to noise ratio (SNR), may also be considered. It was already shown by our group in [17] that degradation of the image, caused mainly through the clutter induced by tissues, could be significant. Obviously, the intensity of this clutter depends on the particular tissues (thickness and type) involved and, as a consequence, an in vivo study will be required to accurately assess these effects on a given body fluid and a given pathology of interest.

Moreover, pathological processes may abnormally increase the concentration of other types of cells such as macrophages or mesothelial cells [24,25], or even significantly change the size distribution of leukocytes themselves [26]. Other components, such as proteins, are often too small in size to reflect a detectable echo, except when their concentration increases as a result of an ongoing disease and clots are formed. All those acoustic scatterers linked to pathological processes may abnormally increase the echogenic properties of the fluid under analysis, but they may also alter the estimation of white blood cell concentration. All these alterations point to the need for a specific analysis of the technique according to the fluid to be analysed and potential pathologies involved. However, even in these cases, abnormal leukocyte results would probably allow the use of the ultrasound estimation of cell concentration as a disease screening technique.

## Figures and Tables

**Figure 1 sensors-23-00977-f001:**
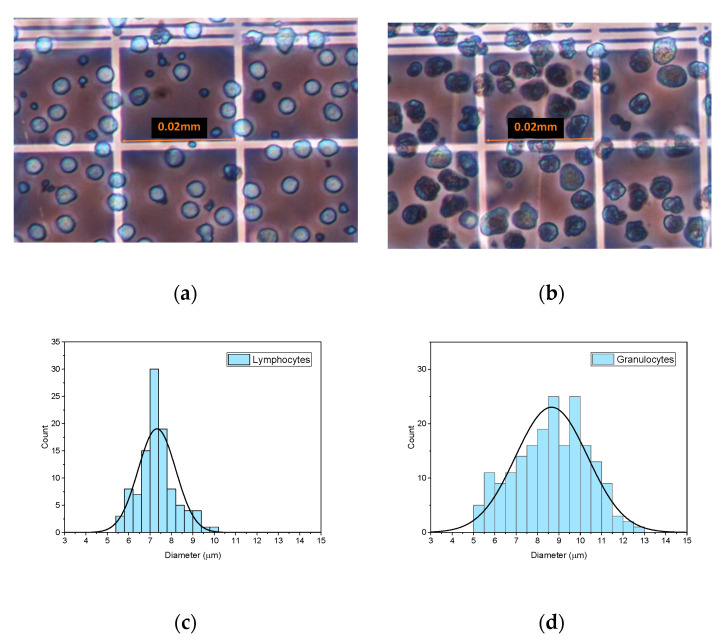
Images of lymphocytes (**a**) and granulocytes (**b**) obtained with the optical microscope from 3000 cells/µL suspensions in the Neubauer haemocytometer. Examples of size diameter histograms corresponding to lymphocytes (**c**) and granulocytes (**d**) with a Gaussian fitting superimposed.

**Figure 2 sensors-23-00977-f002:**
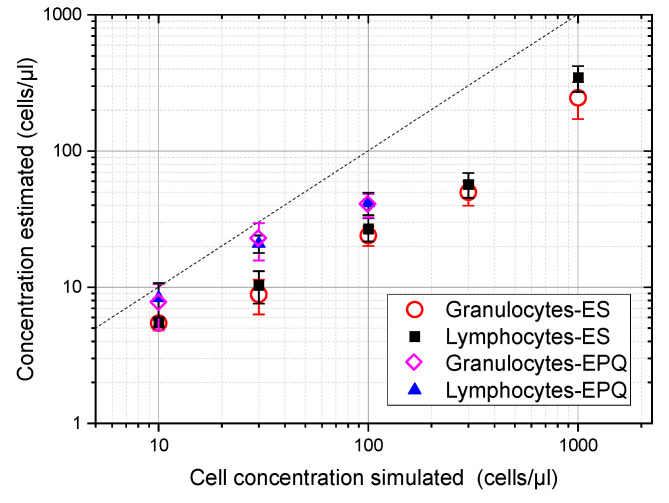
Cell concentration estimation from simulated images of suspensions of lymphocytes and granulocytes using the echo pattern quantification (EPQ) and the envelope statistics (ES) models for single-sized scatterers. The mean sizes and standard deviations of cell suspensions were taken from Table 1. The dash line marks the equivalence between simulated and estimated concentrations.

**Figure 3 sensors-23-00977-f003:**
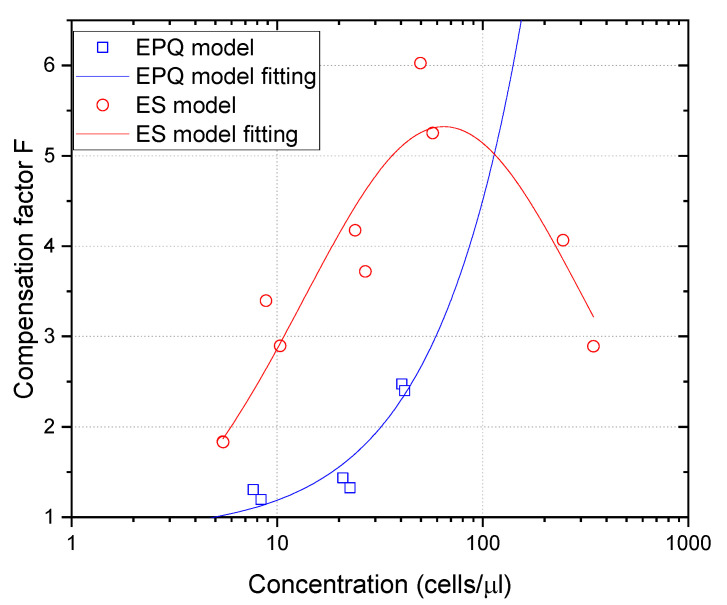
Compensation functions for the EPQ and ES models in blue and red, respectively.

**Figure 4 sensors-23-00977-f004:**
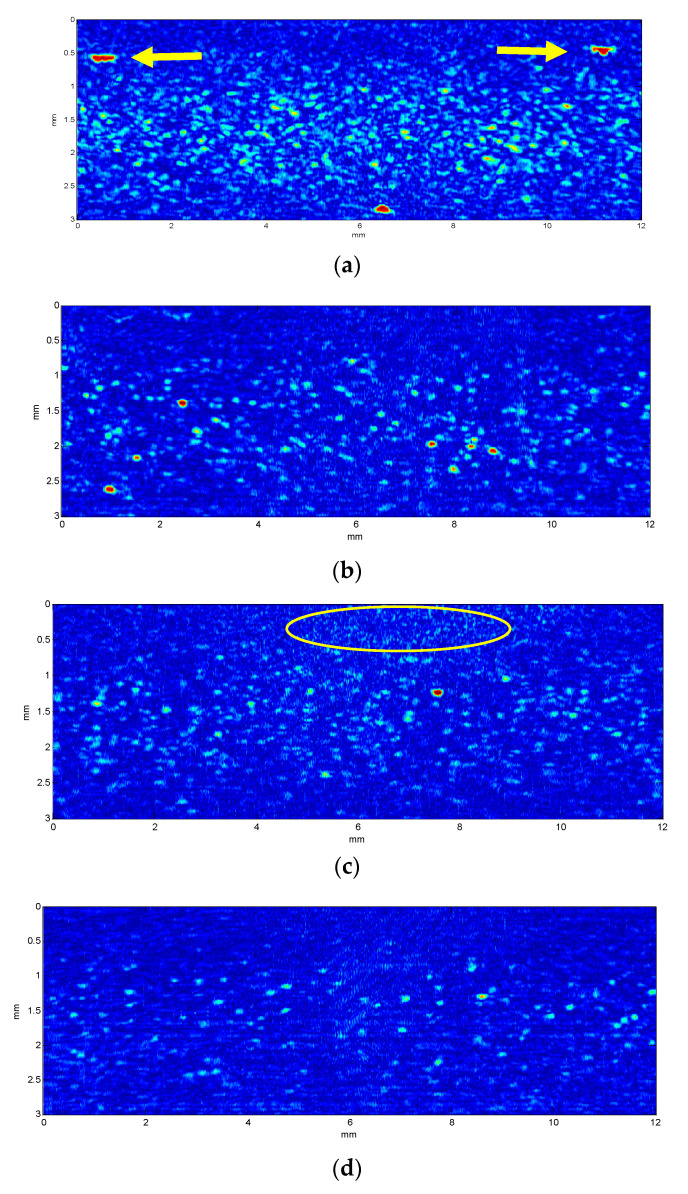
Ultrasound images obtained at 20 MHz for lymphocytes 696 cells/µL (**a**) and 140 cells/µL (**b**), and granulocytes at 165 cells/µL (**c**) and 41 cells/µL (**d**). The yellow arrows in (**a**) point to sample impurities and the circle in (**c**) shows mechanical noise produced by coupling problems between the emitter and the tube sample covered by the polyethylene film.

**Figure 5 sensors-23-00977-f005:**
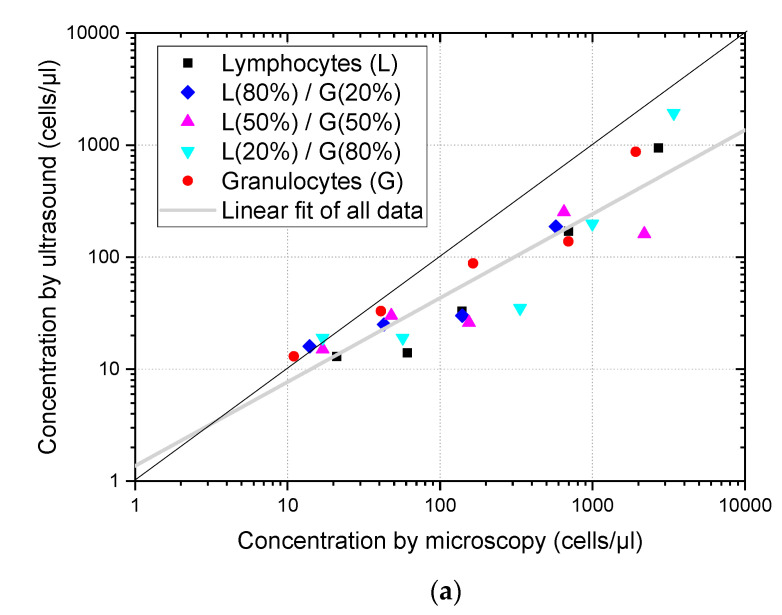
Cell concentration of pure suspensions and mixtures of lymphocytes and granulocytes is estimated from the ultrasound images as a function of the concentration measured using optical microscopy. The diagonal black line corresponds to the concentration coincidence between both methods and the grey line corresponds to the linear fitting of all points. (**a**) Concentration assessment without size dispersion compensation. (**b**) Concentration assessment with size dispersion compensation.

**Table 1 sensors-23-00977-t001:** Mean diameter and standard deviation of lymphocytes and granulocytes.

	Mean Diameter (µm)	Standard Deviation (µm)
Lymphocytes	7.9	1.8
Granulocytes	8.8	2.4

## Data Availability

Data are unavailable due to confidentiality restrictions.

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
