# Peer review of "Evaluation of the Cell Concentration in Suspensions of Human Leukocytes by Ultrasound Imaging: The Influence of Size Dispersion and Cell Type"

_sensors, 2023, doi:10.3390/s23020977_

Round 1

Reviewer 1 Report

The authors present a method for estimating cell concentration using ultrasound. Compared to previous work, their method explores higher concentrations of cells (up to 3000 cells/microL) which is relevant for in vivo applications and diseases such as meningitis. Their method is also independent of ultrasonic amplitude and therefore is more practical in a clinical setting where patient composition varies.

This article is an extension of previous work [10-12], but extends into a higher concentration range where size dispersion becomes important. To compensate for this, they use simulations to estimate the effect of size dispersion on estimates of cell concentration and derive correction factors to compensate for this effect. Subsequently, these correction factors are applied to experimental data. This compensation factor improves the accuracy of the measurements by ~16%. 

This work is an improvement on previous work and substantiates publication in Sensors. 

I recommend the following improvements to the article:

Introduction, sentence #2: For completeness, what pathologies are cell concentrations in urine and serous fluids relevant for?  

Section 2.2.1: references for all percentages should be given 

Discussion: Limitations of this study should be clearly outlined and what future studies must be done to create more in vivo like conditions. For example: 

  • - Pure samples consisting of only leukocytes are used. What are the expected limitations of measurements in whole fluids. E.g. serous fluids contains additional proteins, macrophages, etc. How could one determine cell concentration of only a single component (leukocytes) in these complex fluids in vivo? 

  • - The technique does not theoretically rely on ultrasonic amplitude. However, it would be beneficial to empirically test this. Ultrasound waves will have different SNR and frequency content depending on overlying tissues. An experiment with overlying tissue of various attenuation / scattering strengths would help to confirm the amplitude-independence of this technique and identify the method’s accuracy in a more realistic experiment. 

Reviewer 2 Report

The authors present a new method to compensate the influence of cell size dispersion to achieve reliable human leukocyte concentration estimation from ex vivo ultrasound imaging experiments by taking advantages of correction factors retrieved from numerical simulation and cell size distribution measured by optical microscopy. The authors show that with their cell size dispersion compensation method, lymphocyte concentrations in different cell type mixtures can be accurately measured. The proposed method has its merit per se. However, there are a few key points the authors need to address:

1)    The authors have neglected more powerful non-image-based scatterer concentration measurement concept: dynamic light scattering and its ultrasound counterpart, dynamic ultrasound scattering. These methods allow accurate measurements of scatter size distribution and concentration simultaneously. To me, this is particularly important considering that leukemia patients with certain mutations can have different lymphocyte size distribution (for instance, reference 1). I haven’t extensively searched for diseases that can cause lymphocyte size changes, just to give an example why it matters to be able to measure size and concentration from the same experiment. The authors need to motivate why imaging-based methods should be used at the first place, given that the scatter sizes need to be measured by another technique, for instance, optical microscopy as shown in this study. If typical commercial medical ultrasound machines can be used, then it could be advantageous, especially considering that the authors have a long-term vision to apply their techniques for in vivo measurement on patients. However, based on the description given by the authors, they need a temporal sampling frequency of 100MHz, way higher than ~30 Hz of sampling frequency offered by typical commercial medical ultrasound machines, meaning that they would anyhow need a specialized tool for such measurements. The authors need to motivate better why their proposal of imaging-based solution is worthy readers’ attention and discuss the pros and cons of different methodologies.

Reference 1: Tanahashi T, Sekiguchi N, Matsuda K, Takezawa Y, Ito T, Kobayashi H, Ichikawa N, Nishina S, Senoo N, Sakai H, Nakazawa H, Ishida F. Cell size variations of large granular lymphocyte leukemia: Implication of a small cell subtype of granular lymphocyte leukemia with STAT3 mutations. Leuk Res. 2016 Jun;45:8-13. doi: 10.1016/j.leukres.2016.04.001. Epub 2016 Apr 4.

2)    The authors didn’t write explicitly which plugin or function from ImageJ was used to calculate the cell size distribution in section 2.2.2. More importantly, it is better to provide the cell size distribution histograms of lymphocytes and granulocytes than the mean diameter and standard deviations, as it is unclear whether their sizes follow Gaussian distributions or not. If they don’t, it will be interesting to know to what degree of deviation from Gaussian distribution can their method still yield satisfactory results.  

Some minor points:

3)    In several places, the authors use the word “studio” which I think should be replaced by "study" instead. Please correct them.

4)    In line 380, “big echoes” sounds weird. I would suggest replace “big” by “strong” instead.

Reviewer 3 Report

The given study focuses on using ultrasound imaging to evaluate the cell concentration of dilute leukocyte suspensions in the range (10-3000 cells/μl). The proposed technique is the extension or applied study on leukocytes based on previously published papers [10-11]. The previous studies were based on suspensions of 7 µm and 12 µm particles. The given article is based on leukocytes within the size of  7 µm to 12 µm . However, the authors are required to improve the paper with significant revisions. Here are some of the comments which should be addressed:

1)      Line 97, the word 'studio' seems inappropraite. Similarly, at line 265 and other parts of paper.

2)      Line 114, what do you mean by "0,003–0,06" ? Why authors use commas in the number.

3)      Line 130, authors must define the s, sigma and μ parameters.

4)      Authors have repeatedly use the word "this". Such statments are not clear and need revision, such as:

a)       Line 134, "Once this relation"... what this relationship?

b)      Line 133, "relation between this μ parameter"

5)      Does the testing sample used as blood serum or blood sample?

6)      How did the authors perform simulations? Which platform?

7)      The information about the sample collection center is missing, which country and state?

8)      Line 192, what do you mean by another milliliter sample of PBS? Is it a 1 mL PBS sample?

9)      Specifications about the transducer are missing.

10)   Line 241, the statement "8 acquisitions of 400 samples were averaged" is unclear. What do you mean by averaging data acquisition? The average or mean of signal used for noise reduction. It seems to be inappropriate here.

11)   Figure 1, please check the legend. There is a big mistake in (a) and (b). Which one is lymphocytes and granulocytes?

12)   The echo pattern quantification (EPQ) and envelope statistics (ES) models have been mentioned and cited as previous work. The information about the algorithms is missing in this paper on which the studies are based.

Round 2

Reviewer 2 Report

It is wrong to state that imaging-based method as proposed by the authors can be advantageous than dynamic ultrasound scattering for low cell concentration measurement. The correlation function used in dynamic light /ultrasound scattering or even fluorescence correlation spectroscopy (which I had worked with for many years) guarantees single particle sensitivity as the mean normalized fluctuation (deviation from the mean) which defines the correlation function is inversely proportional to the number of particles. The larger the number of particles in a given focal volume, the smaller fluctuations that can be observed due to diffusions of single particles into and out of the volume. Because of the underlying physical principle, these methods are known to be particularly suitable for measurement of particles at low concentrations. At very high concentrations, I believe the dynamic ultrasound scattering method will require prolonged measurement to get stable correlation curves as short measurements will produce quite noisy data, based on my personal experience on fluorescence correlation spectroscopy. For imaging-based method, this can also be problematic as cells can easily overlap, posing a significant challenge for quantifications. 

The authors suspect that the influence of human body movement can have a larger influence on dynamic ultrasound scattering than ultrasound imaging-based method. This is hardly possible, as such movements can easily introduce motion blur to the acquired ultrasound images making it very hard for analysis. For time correlation-based dynamic scattering methods, as long as the body motion has a different time scale than the cellular motion, these motions can be easily separated and analyzed by fitting to a correlation function involving different time constants. Influence of irregular particle shapes on the correlation curves is a well-studied problem in the field of optics, for which, the authors can search the relevant literature themselves. Nonetheless, the authors have to assume all blood cells to be spherical for their proposed correction method.

I however can agree that imaging helps find the region of interests to perform size and concentration measurements. But for the intended application of the authors, I would definitely choose dynamic ultrasound scattering for in vivo measurements of blood cells and obtain their sizes and concentrations simultaneously instead of using their proposed method that relies on an external method for cellular size estimation in order to measure their concentration. I would like to be clear that I consider their contribution not significant enough for the intended research problem. However, because the publication philosophy in Sensors does not put a strong emphasis on the scientific significance of a study but its robustness if I understand it correctly, I can recommend acceptance of the manuscript under the condition that the authors have a fair introduction and discussion of the methods I bring forward to the attentional sphere of the authors.

Author Response

Please, find the answer in the document attached

Reviewer 3 Report

The authors have carefully revised the major concerns.

Author Response

Authors want to thanks again for the review which helped to improve the manuscript.